# The early-life exposome modulates the effect of polymorphic inversions on DNA methylation

Natàlia Carreras-Gallo [1,19], Alejandro Cáceres [1,2,3,19], Laura Balagué-Dobón [1], Carlos Ruiz-Arenas [4,5,6], Sandra Andrusaityte [7], Ángel Carracedo [8,9], Maribel Casas [1,2,6], Leda Chatzi [10], Regina Grazuleviciene [7], Kristine Bjerve Gutzkow [11], Johanna Lepeule [12], Léa Maitre [1,2,6], Mark Nieuwenhuijsen [1,2,6], Remy Slama [12], Nikos Stratakis [1], Cathrine Thomsen [11], Jose Urquiza [1,2,6], John Wright [13], Tiffany Yang [13], Geòrgia Escaramís [2,14,15], Mariona Bustamante [1,2,6,16], Martine Vrijheid [1,2,6], Luis A. Pérez-Jurado [4,5,6,17] & Juan R. González [1,2,18 ✉]

Polymorphic genomic inversions are chromosomal variants with intrinsic variability that play important roles in evolution, environmental adaptation, and complex traits. We investigated the DNA methylation patterns of three common human inversions, at 8p23.1, 16p11.2, and 17q21.31 in 1,009 blood samples from children from the Human Early Life Exposome (HELIX) project and in 39 prenatal heart tissue samples. We found inversion-state specific methylation patterns within and nearby flanking each inversion region in both datasets. Additionally, numerous inversion-exposure interactions on methylation levels were identified from early-life exposome data comprising 64 exposures. For instance, children homozygous at inv-8p23.1 and higher meat intake were more susceptible to *TDH* hypermethylation ($P = 3.8 \times 10^{-22}$); being the inversion, exposure, and gene known risk factors for adult obesity. Inv-8p23.1 associated hypermethylation of *GATA4* was also detected across numerous exposures. Our data suggests that the pleiotropic influence of inversions during development and lifetime could be substantially mediated by allele-specific methylation patterns which can be modulated by the exposome.

[1] Barcelona Institute for Global Health (ISGlobal), Barcelona, Spain. [2] Centro de Investigación Biomédica en Red en Epidemiología y Salud Pública (CIBERESP), Madrid, Spain. [3] Department of Mathematics, Escola d'Enginyeria de Barcelona Est (EEBE), Universitat Politècnica de Catalunya, Barcelona 08019, Spain. [4] Institut Hospital del Mar d'Investigacions Mediques (IMIM), Barcelona, Spain. [5] Centro de Investigación Biomédica en Red de Enfermedades Raras (CIBERER), Madrid, Spain. [6] Department of Health and Experimental Sciences, Universitat Pompeu Fabra (UPF), Barcelona, Spain. [7] Department of Environmental Science, Vytautas Magnus University, 44248 Kaunas, Lithuania. [8] Medicine Genomics Group, Centro de Investigación Biomédica en Red Enfermedades Raras (CIBERER), University of Santiago de Compostela, CEGEN-PRB3 Santiago de Compostela, Spain. [9] Galician Foundation of Genomic Medicine, Instituto de Investigación Sanitaria de Santiago de Compostela (IDIS), Servicio Gallego de Salud (SERGAS), Santiago de Compostela, Galicia, Spain. [10] Department of Preventive Medicine, Keck School of Medicine, University of Southern California, Los Angeles, CA, USA. [11] Department of Environmental Health, Norwegian Institute of Public Health, 0456 Oslo, Norway. [12] Institut national de la santé et de la recherche médicale (Inserm) and Université Grenoble-Alpes, Institute for Advanced Biosciences (IAB), Team of Environmental Epidemiology applied to Reproduction and Respiratory Health, Grenoble, France. [13] Bradford Institute for Health Research, Bradford Teaching Hospitals NHS Foundation Trust, Bradford, UK. [14] Department of Biomedical Science, Faculty of Medicine and Health Science, University of Barcelona, Barcelona, Spain. [15] Research Group on Statistics, Econometrics and Health (GRECS), UdG, Girona, Spain. [16] Center for Genomic Regulation (CRG), Barcelona Institute of Science and Technology (BIST), Barcelona, Spain. [17] Genetics Service, Hospital del Mar, Barcelona, Spain. [18] Department of Mathematics, Universitat Autònoma de Barcelona, Bellaterra, Spain. [19] These authors contributed equally: Natàlia Carreras-Gallo, Alejandro Cáceres. ✉email: juanr.gonzalez@isglobal.org

nversions are segments of DNA that run in the opposite direction to a reference genome. They are balanced mutations of different sizes, from a gene's exon to a chromosome's portion[1]. Because of their role in adaptation to the environment, chromosome evolution, and sex-determination systems in multiple species, polymorphic inversions have traditionally displayed a great interest in evolutionary biology[2,3]. Recent studies have shown that they are important contributors to the genetic basis of common complex diseases in humans, such as obesity, diabetes, asthma, cancer, and neurological conditions such as depression or neuroticism[4–11]. By capturing multiple functional variants, inversions can confer simultaneous risks to different diseases, and, as such, increase the frequency of the diseases' comorbidities. Human inversions at 8p23.1, 16p11.2, and 17q21.31 are large, common, and associate with multiple diseases, including those co-occurring with obesity[5,8]. In addition, they have been strongly correlated with the expression of the several genes they encapsulate across multiple tissues[8,12–14]. There are different mechanisms from which inversions can modulate gene expression. First, inversions can break genes or displace regulatory elements with important functional and phenotypic consequences[10,12,15]. Second, recombination is suppressed in the inverted region in heterokaryotypes. As such, inverted and noninverted alleles accumulate different genetic variants that support differences of gene expression between alleles[2,16,17]. Although several studies have demonstrated the effect of inversions on gene expression, it is unknown the extent to which inversions are also characterized by specific methylation patterns.

DNA methylation, the addition of a methyl group in a CpG DNA site, plays an important and complex role in the regulation of gene expression[18]. Depending on the relative position of the CpG site within the gene, its methylation can increase or decrease the gene's expression[19]. Methylated promoters are often associated with deactivation of transcription, while methylation within the gene's body avoids alternative start sites[20]. Methylation is often strongly correlated across contiguous CpG sites, a fact that is used to determine differentially methylated regions (DMR) of kilobase-pair lengths[21]. At larger distances, coherent methylation patterns may be supported by genomic variants such as copy number variants[22]. However, it is unknown if methylation patterns in inverted regions can also be detected. We, therefore, hypothesized that the common human inversions at 8p23.1, 16p11.2, and 17q21.31 are correlated with the methylation of multiple CpG sites within and surrounding the inverted region, creating allele-specific methylation patterns. In support of this hypothesis, some studies have already reported associations between inversion and phenotypes likely modulated by specific methylation changes[6,23,24]. Besides, since CpG methylation is involved in regulating chromatin structure[25], these methylation patterns could be associated with different tridimensional (3D) DNA structures for each allele. This would be in line with the influence on 3D DNA structure by large structural variants reported by Shanta et al.[26].

The epigenetic landscape of genes can be altered due to environmental exposures, leading to disease[27–29]. In 2005, Wild introduced the term "exposome" that encompasses all the environmental exposures to which an individual was subjected, from conception to death[30]. This concept has evolved and now it does not only include environmental exposures but also exposures to diet, behavior, and endogenous processes[31]. Common exposures, like air pollution, stress, and heavy metals, among many others, have been associated with distinct epigenetic marks in relevant genes. For example, psychosocial stressors early in life, even in utero, can induce methylation changes on specific genes in the brain[32]. Studies have demonstrated, for instance, that abnormal DNA methylation can lead individuals to be more sensitive to stressful stimuli, increasing the stress burden and anxiety over the life course[33]. More generally, Teh et al. demonstrated that only 25% of the interindividual variation in neonatal DNA methylation was explained by genetic variants, while the 75% was better explained by the interaction of genotype with different *in utero* environments (considering maternal smoking, maternal BMI, and maternal depression, among others)[34]. Therefore, given its strong link with exposome and genetic variation, methylation is currently considered an important target of gene-environment interactions[35].

Here, we first evaluated whether three common polymorphic inversions in humans affect the methylation patterns of their encapsulated and surrounding DNA sequences in blood cells from children and in prenatal heart tissue. Second, using a large set of 64 early-life exposures, we then asked which of these exposures had a different impact on DNA methylation according to the inversion status at 8p23.1, 16p11.2, and 17q21.31.

## Results

**Frequency of inversions at 8p23.1, 16p11.2, and 17q21.31**. We analyzed data from the Human Early Life Exposome (HELIX) project, a multicenter European cohort (Spain, United Kingdom, France, Lithuania, Norway, and Greece). This project comprises 1301 children with genomic, transcriptomic, epigenomic, and exposome data[36]. HELIX has the goal of characterizing the exposome during early life and evaluating its relationship with molecular signatures and child health outcomes. The genome-wide blood DNA methylation and blood cell transcriptome were measured at the ages between 6 and 11. From this dataset, we selected children with genetic and methylation data. We used Peddy[37] to estimate major population ancestry groups and individuals of European ancestry were kept in the analysis, resulting in a total of 1009 children included in the analyses.

We called 8p23.1, 16p11.2, and 17q21.31 inversion genotypes from the selected children using *scoreInvHap*[11] on imputed SNP array data. Inversion genotypes were labeled as N/N for noninverted homozygous, N/I for heterozygous, and I/I for inverted homozygous. We observed that the frequencies for the inverted allele were consistent with those reported for Europeans (55.70%, 35.70%, and 21.95% for inversions at 8p23.1, 16p11.2, and 17q21.31, respectively)[1,11]. As expected, we did not observe significant variation between sexes (Supplementary Fig. 1a–c), but we observed some variations across cohorts (Supplementary Fig. 1d–f). As previously reported[8], we evaluated the south–north gradient for the inverted allele frequency and we observed a positive correlation for inv-16p11.2 ($r = 0.79$, $P = 0.058$), and a negative correlation for inv-17q21.31 ($r = -0.92$, $P = 0.009$) (Supplementary Fig. 2). For the inv-8p23.1, we did not observe a significant south–north gradient ($r = -0.33$, $P = 0.519$).

**Inversions as *eQTLs* in blood cells**. We first evaluated the inversion status as expression quantitative trait loci (*eQTL*) of the genes within the inversion regions ±1 Mb. We performed the association analyses of the inversions in each separate cohort adjusting by sex, age, cell-type proportions (inferred from methylation data), and 10 genome-wide principal components of genomic SNP variation ($N = 790$). We then combined the results with a meta-analysis across cohorts. The results were considered significant when they passed Bonferroni's correction for multiple comparisons. We confirmed that the inv-8p23.1 and inv-16p11.2 were *eQTLs* for the numerous neighboring genes and the genes they encapsulate (see Supplementary Data 1 and Supplementary Fig. 3). We observed 12 genes that were significantly associated with inv-8p23.1. We detected significant upregulation of *BLK*, *SLC35G5/SLC35G4, FAM86B1/FAM86B2*, and *FAM86B3P*, and

downregulation of *FDFT1, FAM167A, FAM66D, SGK223, XKR6*, and *LOC100506990* for the inverted allele. In the case of the polymorphic inversion at 16p11.2, we observed 10 significant associations, including upregulation of *TUFM, MIR4721, EIF3C/EIF3CL, LAT, SPNS1*, and *NPIPB9/NPIPB8/NPIPB7* for the inverted allele and downregulation of *SGF29, SBK1, LOC388242*, and *SULT1A1*. Finally, for inv-17q21.31, we did not observe *eQTL* effects, perhaps because single-copy genes within this inversion are mostly expressed in the brain[14]. We thus confirmed the effect of the inversions 8p23.1 and 16p11.2 on the gene expression in blood in 6–11-year-old children, as previously observed in adults across different tissues[8,12–14].

**Inversions as *mQTLs* in blood cells**. We then studied the associations of the genotypes of each of the three inversions with the differential methylation of CpG sites within the ±1-Mb regions containing the inversions (Supplementary Data 2). We removed CpG sites with single-nucleotide polymorphic (SNP) variation. We performed the analyses in each separate cohort adjusting by the same covariates likewise the transcription analyses. We combined the results with a meta-analysis across cohorts ($N = 1009$). As illustrated in Fig. 1a–c, all three inversions were significantly associated with differences in methylation across multiple CpG sites after Bonferroni's correction for multiple comparisons. We also observed that the most significant associations were in CpG sites within the inversion region or close to the breakpoints. In particular, we observed that 15.21% (129 of 848) CpG sites within and around inv-8p23.1 had significant differences in methylation levels according to to the inversion status (min. $P = 63.1 \times 10^{-147}$, Fig. 1a), with 49 significant CpG sites hypermethylated and 80 hypomethylated in the inverted concerting the noninverted allele. For this inversion, we observed 24 genes with at least one significant differentially methylated CpG site and five genes with more than five differentially methylated sites; namely *MSRA, MFHAS1, BLK, RP1L1*, and *XKR6*. For inv-16p11.2, we found 27 significant CpG sites differentially methylated from a total of 401 (6.73%, min. $P < 10^{-300}$, Fig. 1b), with 9 significant CpG sites hypermethylated and 18 hypomethylated at the inverted allele. For this inversion, we observed 11 genes with at least one significant CpG site. *IL27* was the gene with the greatest number of CpG sites (5) differentially methylated (all hypomethylated at the inverted allele). Finally, 58 CpG sites from 666 (8.71%, min. $P < 10^{-300}$, Fig. 1c) had significant methylation differences for inv-17q21.31 (30 hypermethylated and 28 hypomethylated at the inverted allele). *CRHR1, MAPT*, and *KANSL1* were the 17q21.31 genes with the highest number of differentially methylated CpG sites and a total of 14 genes had at least one CpG site differentially methylated. Therefore, each of these three inversions behaves as an extended methylation quantitative trait loci (*mQTL*) covering hundreds of kilobases, an observation that had not been previously reported.

To establish the degree to which the association between the effect of inversion status on CpG methylation is associated with changes in gene expression of surrounding genes, we searched for the methylation changes that locate in differentially expressed genes (Supplementary Fig. 4). We observed that four genes (*BLK, FDFT1, XKR6*, and *FAM167A*) overlapped for the inv-8p23.1 with differentially methylated CpG sites. We analyzed whether the observed expression changes were in the expected directions based on the methylation of these regions, that is, hypermethylation of the promoters for downregulated genes, hypomethylation of the promoters for upregulated genes, and hypermethylation of the bodies for upregulated genes. *XKR6* was a highly consistent case whose downregulation and methylation, across 11 CpG sites within its body, were associated with the inverted allele. For inv-

16p11.2, we observed four genes that were differentially expressed and methylated by the inversion allele (*TUFM, SBK1, SPNS1*, and *SULT1A1*). In this case, most of the CpG sites were in the promoter region (TSS1500) and the relation between the expression and methylation levels was consistent. We further observed that *SULT1A1* and *TUFM* had CpG sites in their promoters (cg01378222 and cg00348858) that highly associated with the effect of inversion in gene expression. We found that cg01378222 mediated the 95% of the association between inv-16p11.2 and the expression of *SULT1A1* ($P < 2 \times 10^{-16}$), and that cg00348858 mediated the 5% of the association between the inversion and *TUFM* expression ($P = 0.002$).

These findings provided evidence of regulatory pathways where inversion, methylation, and gene expression are all involved. In addition, our observation that inv-17q21.31 did not show *eQTL* effects in blood indicates that the three-way association of the variables is tissue specific, as we observed a clear methylation pattern for the inversion.

**Inversion-state-specific methylation patterns**. In order to define whether the methylation patterns were specific to each inversion allele, we performed principal component (PC) analysis of the methylation levels of CpG sites within and around each inversion. We thus quantified individual differences in methylation profiles across the inverted regions. We included the region ±1 Mb to account for the effect of the inversions beyond the breakpoints. Remarkably, the first component strongly correlated with the inversion genotype of the individuals in all three inversions (inv-8p23.1 PC 1: $R^2 = 0.68$, $P < 2 \times 10^{-16}$, inv-16p11.2 PC 1: $R^2 = 0.05$, $P = 1.34 \times 10^{-12}$, and inv-17q21.31 PC 1: $R^2 = 0.70$, $P < 2 \times 10^{-16}$), see Fig. 1d–f. We observed that the first PC clearly separated the genotypes of inversions at 8p23.1 and 17q21.31, possibly sustained by the haplotypic differences between inversion status. While the first PC of inv-16p11.2 was significantly associated with inversion genotypes, the second PC was also needed to distinctly separate the genotypes ($R^2 = 0.33$, $P < 2 \times 10^{-16}$). This is in line with the univariate differential analysis, where inv-16p23.1 showed the smallest proportion of CpG sites differentially methylated according to the inversion status. This is possibly explained by the multiple haplotypes supported by this inversion[11]. These analyses showed that hyper- and hypomethylation patterns of CpG sites across the inverted regions are specific to the inversion status.

**Inversions as *mQTLs* in fetal heart DNA**. We asked whether the effect of the inversion on DNA methylation could be also seen prenatally and in another tissue. Using methylation data of heart DNA from 39 fetuses from interrupted pregnancies at 21–22 weeks of gestational age due to congenital heart defects[38], we performed the same differential analysis adjusting by sex. We observed that all the inversions act as *mQTLs* during early development from conception, although few CpG sites per inversion passed Bonferroni's threshold (Fig. 1g–i and Supplementary Data 3). This can be explained by the small sample size. Nonetheless, we observed that the distribution of the significant associations was very similar to the one observed in HELIX data, having greater differences in methylation in the CpG sites between the breakpoints. In addition, we saw that 38 CpG significant sites overlapped between heart (nominal *P*-value) and blood (adjusted *P*-value) tissues, 32 of which were in the same direction, suggesting that the effect of inversions on CpG methylation may be sustained between tissues and stages of life.

**Effect of inversion-exposure interactions on DNA methylation**. As these common human inversions at 8p23.1, 16p11.2, and

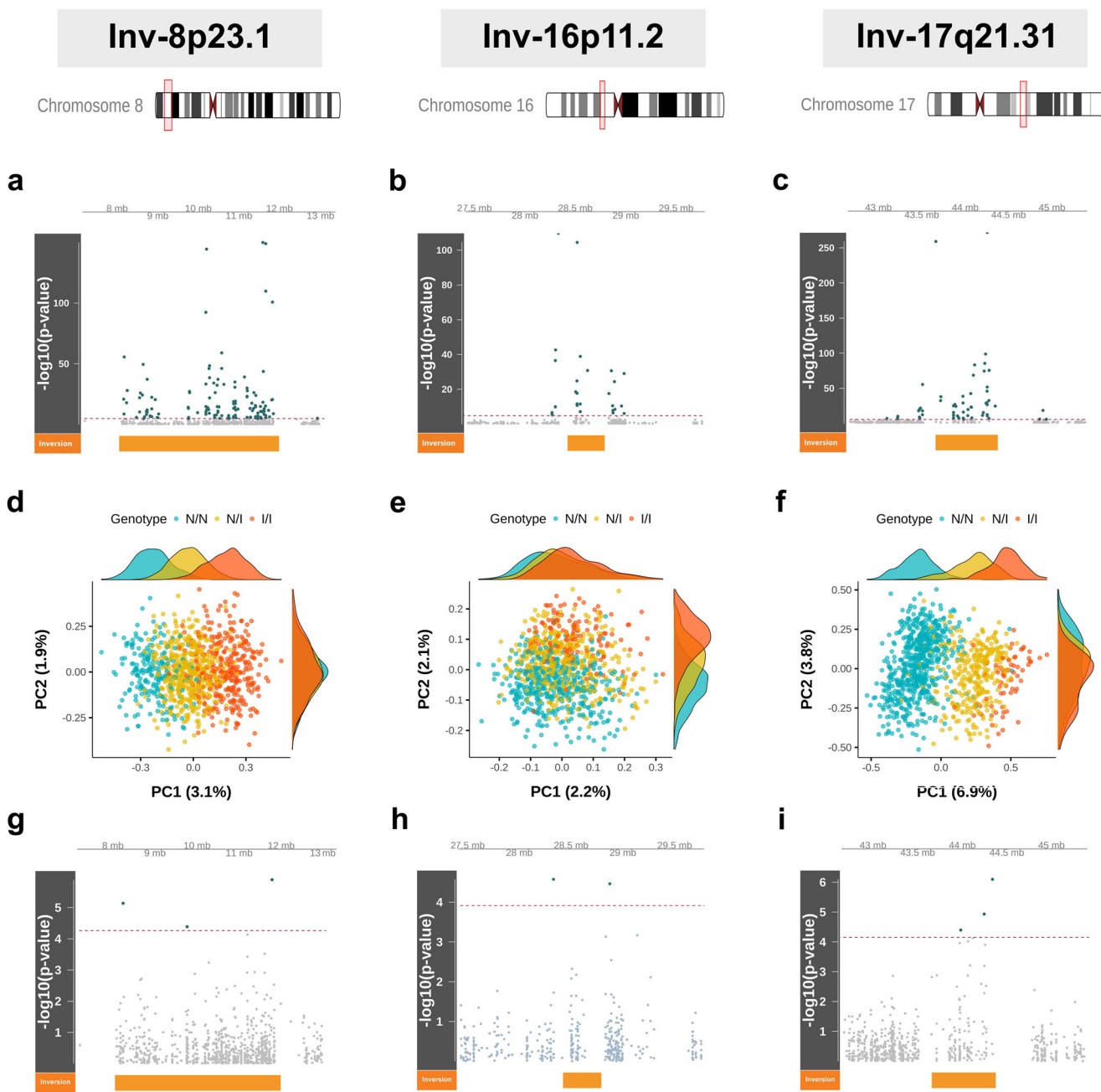

**Fig. 1 Inversion status as methylation quantitative trait loci (*mQTL*) of multiple CpG sites within and surrounding three common human inversions.**
The first column in the plot panel corresponds to inv-8p23.1, the second to inv-16p11.2, and the third to inv-17q21.31. **a–c** Manhattan plots for the
significance of the associations between the differential methylation of the CpG sites and the inversion genotypes in child blood cells (*N* = 1009). The x
axes show the chromosome position (±1 Mb between the inversions' breakpoints). The y axes show the –log₁₀ (*P*-value). The dashed red line indicates
Bonferroni's threshold of significance. Green points are CpG sites with significant associations and those in gray are nonsignificant. The orange block
illustrates the inversions' region. **d–f** Principal component (PC) analysis for methylation levels of CpG sites within and surrounding the inversions, revealing
remarkably distinctive methylation patterns among the different inversion statuses. Blue points illustrate noninverted homozygous (N/N), yellow illustrates
heterozygous (N/I), and orange illustrates inverted homozygous (I/I) individuals. In parenthesis, the methylation variance explained by each PC.
**g–i** Manhattan plots of differentially methylated CpG sites, depending on the inversion genotypes in fetal heart DNA (*N* = 40).

17q21.31 offered a solid genetic context where allele-specific methylation patterns were found, we then asked whether these patterns were modulated by environmental exposures. Thus, we assessed which of 64 exposures at early life differentially modified the methylation levels of the CpG sites within the inversion regions according to the inversion status.

We performed differential methylation analyses for the interactions of the 3 inversions with 64 exposures (7 during pregnancy and 57 at 6–11 years of age) grouped by 12 exposure families, including build environment, air pollution, persistent and nonpersistent chemicals, diet, and exposure to tobacco smoke, among others (Fig. 2a and Supplementary Data 4). We observed 36 exposures and 58 CpG sites implicated in at least one significant inversion-exposure interaction after Bonferroni's correction for multiple comparisons (see Table 1 and Supplementary Data 5). All exposure families had at least one exposure

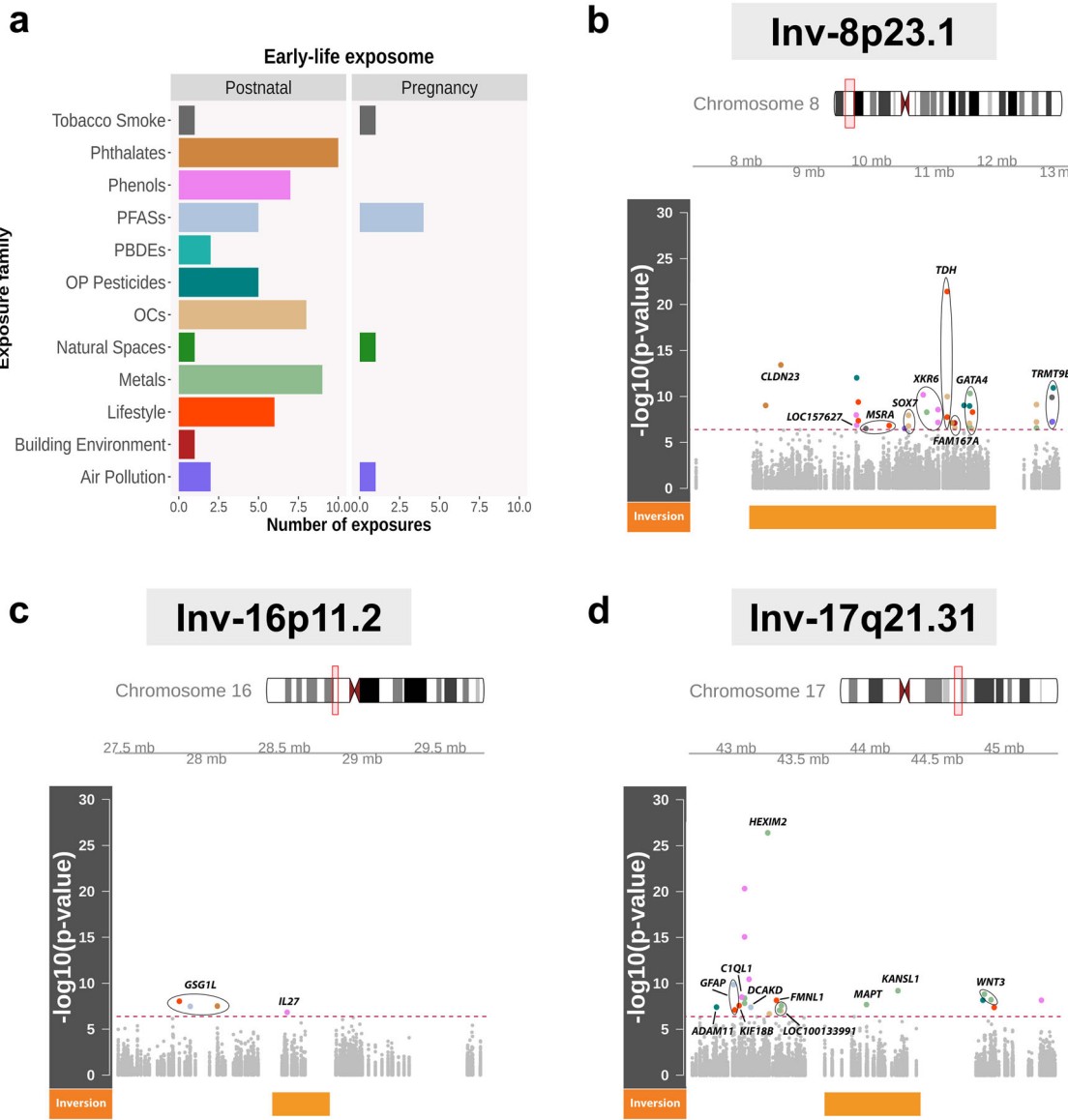

**Fig. 2 Inversion-exposure interactions as methylation quantitative trait loci (mQTL) of multiple CpG sites within and surrounding three common human inversions. a** Number of exposures per family in the early-life exposome from the HELIX project. **b**–**d** Manhattan plots showing the significance of the associations ($N = 1009$) between the differential methylation of the CpG sites and the inversion-exposure interactions across all 64 exposures in (**a**) and the genotypes of three human inversions at 8p.23.1 (**b**); 16p11.2 (**c**); and 17q21.31 (**d**), illustrated by the orange block. The x axes show the chromosome position (±1 Mb between the inversions' breakpoints). The y axes show the –$\log_{10}$ (P-value) of the associations. The dashed red line indicates Bonferroni's threshold of significance. Significant results are colored according to the family exposure (**a**) and labeled according to the closest gene to the CpG (Illumina annotation). Gray points are not significant.

that interacted with one of the three inversions, except natural spaces and polybrominated diphenyl ether compounds (PBDE). Remarkably, the exposure families with the greatest number of significant interactions were metals (13 interactions), diet (11), phenols (11), and organochlorines (OCs) (10) (Supplementary Data 6).

Inversion at 8p23.1 had 36 significant interactions with exposures from 9 different families (Fig. 2b). OC was the most predominant exposure family involved in 8 interactions, followed by diet with 6 and phenols with 5. The genes with the greatest number of CpG sites differentially methylated according to the interactions were *GATA4* (hypomethylated for the inverted allele in all but one), *XKR6* (hypermethylated for the inverted allele in all but one), *TDH*, and *FAM167A*, all of them seen differentially methylated, depending on the inversion haplotype. In the case of

inv-16p11.2, we only found 4 significant interactions (Fig. 2c). Notably, 3 interactions contributed to *GSG1L* methylation changes: child vegetable intake (cg08755784, $\beta = 0.006$, $P = 8.9 \times 10^{-9}$), child mono-2-ethylhexyl phthalate (MEHP) levels (cg03962082; $\beta = -0.011$, $P = 3.0 \times 10^{-8}$), and child perfluorohexane sulfonate (PFHXS) levels (cg01896119; $\beta = -0.014$, $P = 3.3 \times 10^{-8}$). For inv-17q21.31, we observed 24 significant interactions with exposures from 6 exposure families (Fig. 2d). The most frequent family was metals with 9 significant interactions with inv-17q21.31. The most significant interaction of the inversion was with the exposure to lead on *HEXIM2* methylation (cg19655070: $\beta = -0.043$, $P = 4.5 \times 10^{-27}$). Furthermore, several CpG sites in the upstream region of *C1QL1* were differentially methylated according to the interaction of inv-17q21.31 with phenols. In particular, a CpG site within *C1QL1* promoter was

**Table 1 Significant associations between CpG methylation levels and inversion-exposure interactions.**

| Exposure | Exposure family | Period | Inversion | CpG | Location | Gene symbol | Effect | P-value |
|---|---|---|---|---|---|---|---|---|
| Lead | Metals | Postnatal | 17q21.31 | cg19655070 | chr17:43237981 | HEXIM2 | −0.043 | 4.5E-27 |
| Meat intake | Diet | Postnatal | 8p23.1 | cg01489256 | chr8:11204017 | TDH | 0.0156 | 3.8E-22 |
| MEPA | Phenols | Postnatal | 17q21.31 | cg06368300 | chr17:43065840 | | 0.0077 | 5.1E-21 |
| MEPA | Phenols | Postnatal | 17q21.31 | cg11178337 | chr17:43065745 | | 0.0189 | 9E-16 |
| MBzP | Phthalates | Postnatal | 8p23.1 | cg0667706 | chr8:8559999 | CLDN23 | 0.0173 | 3.8E-14 |
| DETP | OP Pesticides | Postnatal | 8p23.1 | cg17526103 | chr8:9765691 | | 0.0038 | 9.5E-13 |
| DMTP | OP Pesticides | Postnatal | 8p23.1 | cg17120402 | chr8:12891262 | | 0.0065 | 1.2E-11 |
| MEPA | Phenols | Postnatal | 17q21.31 | cg07822074 | chr17:43098904 | | 0.0049 | 3.6E-11 |
| Manganese | Metals | Postnatal | 8p23.1 | cg26020513 | chr8:11568356 | GATA4 | −0.033 | 4.8E-11 |
| OXBE | Phenols | Postnatal | 8p23.1 | cg20858107 | chr8:10823238 | XKR6 | −0.004 | 6.7E-11 |
| HCB | OCs | Postnatal | 8p23.1 | cg03399933 | chr8:11205972 | TDH | −0.023 | 1.1E-10 |
| Parental smoking | Tobacco Smoke | Postnatal | 8p23.1 | cg08196601 | chr8:12869553 | TRMT9B | −0.01 | 1.3E-10 |
| PFUNDA | PFASs | Postnatal | 17q21.31 | cg23016243 | chr17:42983768 | GFAP | −0.004 | 1.3E-10 |
| KIDMED score | Diet | Postnatal | 8p23.1 | cg19352062 | chr8:9791449 | | 0.0054 | 4.1E-10 |
| Molybdenum | Metals | Postnatal | 17q21.31 | cg13465858 | chr17:44204908 | KANSL1 | 0.0217 | 6.3E-10 |
| PCB 180 | OCs | Postnatal | 8p23.1 | cg19931644 | chr8:12623485 | | 0.0185 | 7.9E-10 |
| DMDTP | OP Pesticides | Postnatal | 8p23.1 | cg07291889 | chr8:11471712 | | −0.014 | 9.6E-10 |
| MBzP | Phthalates | Postnatal | 8p23.1 | cg19996406 | chr8:8318774 | | −0.008 | 9.7E-10 |
| DEP | OP Pesticides | Postnatal | 8p23.1 | cg22320962 | chr8:11560299 | GATA4 | −0.005 | 1.1E-09 |
| Molybdenum | Metals | Postnatal | 17q21.31 | cg16677019 | chr17:44847268 | WNT3 | −0.2 | 1.5E-09 |
| ETPA | Phenols | Postnatal | 8p23.1 | cg11051055 | chr8:11058145 | XKR6 | 0.0076 | 2.8E-09 |
| ETPA | Phenols | Postnatal | 17q21.31 | cg24945657 | chr17:43044484 | CIQL1 | −0.011 | 3.2E-09 |
| Arsenic | Metals | Postnatal | 17q21.31 | cg06368300 | chr17:43065840 | | 0.0077 | 4.1E-09 |
| KIDMED score | Diet | Postnatal | 8p23.1 | cg12395012 | chr8:11607386 | GATA4 | −0.004 | 5.1E-09 |
| Cadmium | Metals | Postnatal | 8p23.1 | cg02569740 | chr8:10878898 | XKR6 | 0.0093 | 5.2E-09 |
| Mercury | Metals | Postnatal | 17q21.31 | cg16440629 | chr17:44896147 | WNT3 | 0.0073 | 6E-09 |
| DEP | OP Pesticides | Postnatal | 17q21.31 | cg23968286 | chr17:44835681 | | −0.004 | 6.7E-09 |
| OXBE | Phenols | Postnatal | 17q21.31 | cg07673979 | chr17:45270216 | | −0.003 | 6.9E-09 |
| KIDMED score | Diet | Postnatal | 17q21.31 | cg09264140 | chr17:43302776 | FMNL1 | −0.005 | 7E-09 |
| Vegetables intake | Diet | Postnatal | 16p11.2 | cg08755784 | chr16:27829728 | GSG1L | 0.0065 | 8.9E-09 |
| ETPA | Phenols | Postnatal | 8p23.1 | cg01454752 | chr8:9758847 | LOC157627 | 0.0078 | 1.1E-08 |
| HCB | OCs | Postnatal | 8p23.1 | cg24690731 | chr8:10589093 | SOX7 | −0.02 | 1.1E-08 |
| Cobalt | Metals | Postnatal | 17q21.31 | cg06368300 | chr17:43065840 | | −0.022 | 1.4E-08 |
| Meat intake | Diet | Postnatal | 8p23.1 | cg02601489 | chr8:11203954 | TDH | 0.0092 | 1.8E-08 |
| Copper | Metals | Postnatal | 17q21.31 | cg05301556 | chr17:43339594 | MAPT; LOC100128977 | 0.0522 | 2E-08 |
| Cobalt | Metals | Postnatal | 17q21.31 | cg26742995 | chr17:43025343 | LOC100133991; SPATA32 | 0.0198 | 2.6E-08 |
| KIDMED score | Diet | Postnatal | 17q21.31 | cg00240569 | chr17:43025343 | KIF18B | 0.0052 | 2.6E-08 |
| MEHP | Phthalates | Postnatal | 16p11.2 | cg03962082 | chr16:28072873 | GSG1L | −0.01 | 3E-08 |
| PFHXS | PFASs | Pregnancy | 16p11.2 | cg01896119 | chr16:27899404 | GSG1L | −0.014 | 3.3E-08 |
| DMTP | OP Pesticides | Postnatal | 17q21.31 | cg11640208 | chr17:42857157 | ADAM11 | −0.006 | 3.8E-08 |
| PFUNDA | PFASs | Postnatal | 17q21.31 | cg18176312 | chr17:4311632 | DCAKD | −0.006 | 4E-08 |
| Fish and seafood intake | Diet | Postnatal | 17q21.31 | cg17101843 | chr17:44919554 | | −0.01 | 4.1E-08 |
| Vegetables intake | Diet | Postnatal | 8p23.1 | cg00056202 | chr8:9791350 | | 0.0085 | 4.4E-08 |
| PM2.5 (preg) | Air pollution | Pregnancy | 8p23.1 | cg26339990 | chr8:12878608 | TRMT9B | −0.003 | 5.5E-08 |
| Active smoking (preg) | Tobacco smoke | Pregnancy | 8p23.1 | cg08196601 | chr8:12869553 | TRMT9B | −0.02 | 5.9E-08 |

The table illustrates the top 45 significant associations of CpG sites (±1 Mb) and the interactions of three common human inversions (inv-8p23.1, inv-16p11.2 and inv-17q21.31) with exposures in the HELIX exposomic data. The full table is available in Supplementary Data 5. The first column indicates the exposure involved in the interaction (the description of the exposures is detailed in Supplementary Data 4). Exposures are described by their families and the period which they were measured. The inversion column describes the inversion interacting with the exposure. CpG sites are described by their name, location, and gene symbol (written in italics), when mapped to a gene. The Effect column represents the estimate of the interaction effect and the P-value column its nominal level of significance.

hypomethylated for the inverted allele when the ethyl paraben (ETPA) exposure increased (cg24945657: $\beta = -0.011$, $P = 3.2 \times 10^{-9}$). In addition, three intergenic CpG sites near this gene promoter were hypermethylated for the inverted allele when the exposure to methyl paraben (MEPA) increased (cg06368300: $\beta = 0.008$, $P = 5.1 \times 10^{-21}$; cg11178337: $\beta = 0.019$, $P = 9.0 \times 10^{-16}$; cg07822074: $\beta = 0.005$, $P = 3.6 \times 10^{-11}$). It should be noted that there are four genes (KANSL1, MAT, LOC100128977, and WNT3) in this region with significant associations that were also differentially methylated, depending on the inversion haplotype.

**Genes with the strongest and most numerous inversion-exposure interactions.** Within the significant interactions (Table 1), we looked in detail at the genes that showed both the highest significant levels and multiple interactions across different CpG sites for the same gene. We identified three relevant genes within inv-8p23.1, namely TDH, GATA4, and TRMT9B. Within TDH, we found two CpG sites significantly associated with the interaction between the inversion and meat intake: cg01489256 ($\beta = 0.0156$, $P = 3.8 \times 10^{-22}$) and cg02601489 ($\beta = 0.0092$, $P = 1.8 \times 10^{-8}$). More specifically, we observed that individuals homozygous for the noninverted allele (N/N) had a negative association, while heterozygous individuals did not present any association, and homozygous for the inverted allele (I/I) had a positive association (Fig. 3a). We also observed that the association was consistent across all the cohorts, with no significant heterogeneity (cg01489256: $P = 0.39$; cg02601489: $P = 0.45$), see Fig. 3b. We further observed that the increase of meat intake reduced the expression of TDH ($P = 0.00398$), while the associated methylation effect on the expression depended on the genetic context given by the inversion, adjusting by sex, age, and cohort (CpG-inversion interaction, $P = 0.00193$) (Supplementary Fig. 5). Remarkably, the gene, the inversion, and the exposure have been independently associated with obesity in adults[5,39–41].

GATA4 was the gene with the greatest number of CpG sites that changed their methylation according to different interactions between inv-8p23.1 and exposures from different families. These interactions included manganese (cg26020513: $\beta = -0.033$, $P = 4.8 \times 10^{-11}$), diethylphosphate (DEP) (cg22320962: $\beta = -0.005$, $P = 1.1 \times 10^{-9}$), Mediterranean Diet Quality Index for children and teenagers (KIDMED) (cg12395012: $\beta = -0.004$, $P = 5.1 \times 10^{-9}$), mercury (cg27100236: $\beta = -0.007$, $P = 1.8 \times 10^{-7}$), and PCB 138 (cg13293535: $\beta = 0.013$, $P = 3.5 \times 10^{-7}$) exposures. We observed that this CpG was hypermethylated in the individuals homozygous for noninverted allele when increasing the exposure to manganese (Fig. 3c). The meta-analysis also revealed consistency across cohorts with no significant heterogeneity ($P = 0.74$) (Fig. 3d). Interestingly, hypermethylation of GATA4 in developing heart DNA, particularly at cg26020513, has been previously associated with congenital heart defects in fetuses[42].

Another interesting result of our analysis relates to the methylation of the TRMT9B gene, also known as C8orf79 or KIAA1456, a tRNA methyltransferase. The gene has been seen to associate with laryngotracheitis, an upper respiratory tract disease in chicken[43,44]. We observed that parental smoking during childhood significantly modulated the inversion-associated methylation of cg08196601 ($\beta = -0.010$, $P = 1.3 \times 10^{-10}$) (Fig. 3e). The interaction of the inversion with maternal smoking during pregnancy was also associated with the methylation of cg08196601 ($\beta = -0.020$, $P = 5.9 \times 10^{-8}$). In addition, the methylation of cg26339990 was associated with the interaction of the inversion with outdoor PM2.5 (an air pollution exposure) during pregnancy ($\beta = -0.003$, $P = 5.5 \times 10^{-8}$). In the three cases, the noninverted allele was associated with increased levels of methylation with the exposures. We observed that the heterogeneity across cohorts was not significant ($P = 0.63$) (Fig. 3f). In line with these observations, the noninverted allele for inv-8p23.1 has been found to associate with asthma[5] while parental smoking and exposure to high levels of PM2.5 during pregnancy or childhood increase the risk of respiratory diseases in children[45–47].

## Discussion

Here, we show that the common human chromosomal inversions at 8p23.1, 16p11.2, and 17q21.31 have distinctive methylation patterns in blood across the inverted regions and that the early-life exposome modulates these patterns. We observed that during childhood, approximately 10% of the CpG sites within the inverted regions ±1 Mb were significantly differentially methylated according to the inversion genotype. The amount of the differentially methylated CpG sites was high within the region and sharply decreased after the breakpoints, indicating the targeted effect of genomic inversions on DNA methylation. We could also identify the effects of the inversions at prenatal stages in heart tissue, suggesting their relevant role during development even in utero. As such, inversions are early methylation quantitative loci for the genes they enclose. Our findings, therefore, add to other effects that inversions have on gene expression[8,13,14,48], derived from their genetic variability or from the displacement of regulatory elements near the breakpoints[10]. While individual CpG associations with the inversion may be due to the inversion or to local genetic variability in linkage with the inversion, our observations in the PC analysis reveal a spatial pattern given by the correlation of several CpG-site associations that fits the extension of the inversion. It is clear that the cause of such extended pattern along the affected sequence has been produced by the presence of the inversion, likely due to both the DNA reconfiguration and the accumulation of specific genetic variability along the segment that results from the suppression of recombination between inversion states.

We show that an important influence of inversions on phenotypes could be derived from the methylation patterns they support. Few previous studies have analyzed targeted methylation changes when studying a specific inversion or disease. We previously reported that the effect of inv-17q21.31 on colorectal disease-free survival is more likely mediated by DNA methylation than by gene expression[6]. Here, we document that the effect of inversions on methylation is strong along the inverted segment and already significant during early embryonic and fetal development in heart-tissue DNA. One of the main established mechanisms underlying the influence of inversions on phenotypic traits and their pleiotropy is the suppression of recombination within the inverted sequence in heterozygotes. Allele combinations can thus be protected, leading to the generation and possible selection of specific haplotypes for each inversion state[10]. In addition, inversion breakpoints can disrupt coding regions or regulatory elements, altering gene expression or generating novel transcripts with phenotypic consequences, including deleterious effects[15]. These effects likely play a role in the association of these three polymorphic inversions with complex diseases, like obesity[5,8], autoimmune diseases[49], or neurodegenerative disorders[50–52]. For these diseases with important environmental components, our results further suggest the additional role of inversion-associated methylation that is modifiable by environmental exposures.

Allele-specific methylation patterns in inversions can be caused or facilitated by their specific genetic variability and/or different chromatin structure. In our study, we removed probes with SNPs

## cg01489256 (*TDH*) - Meat Intake * Inv-8p23.1

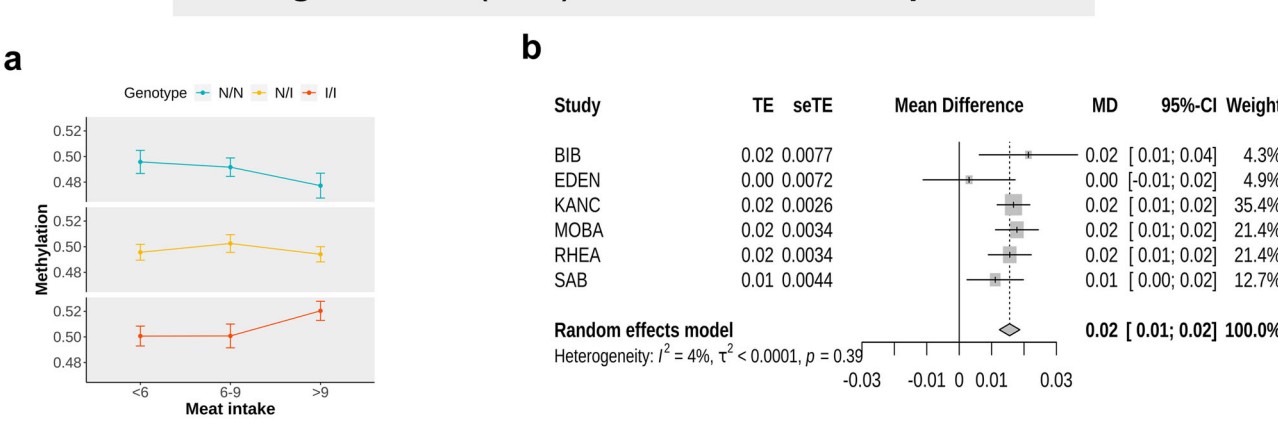

## cg26020513 (*GATA4*) - Manganese * Inv-8p23.1

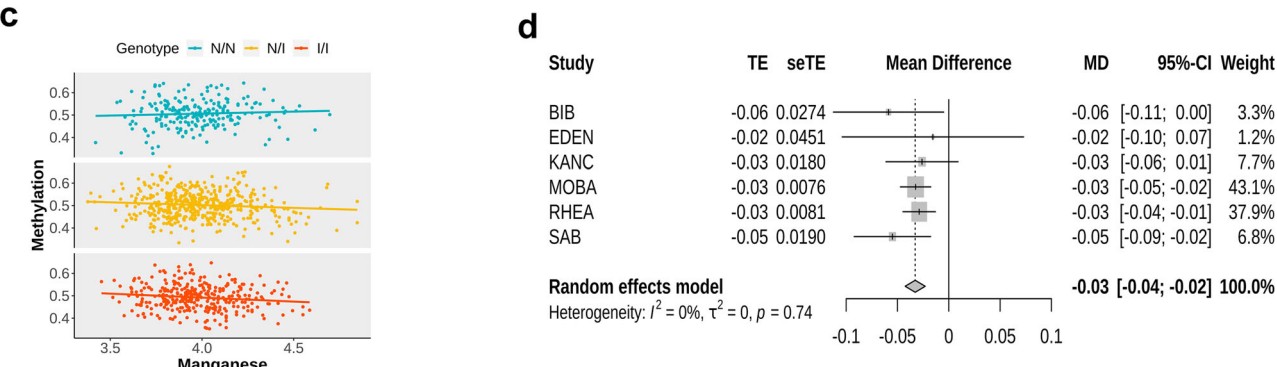

## cg08196601 (*TRMT9B*) - Parental smoking * Inv-8p23.1

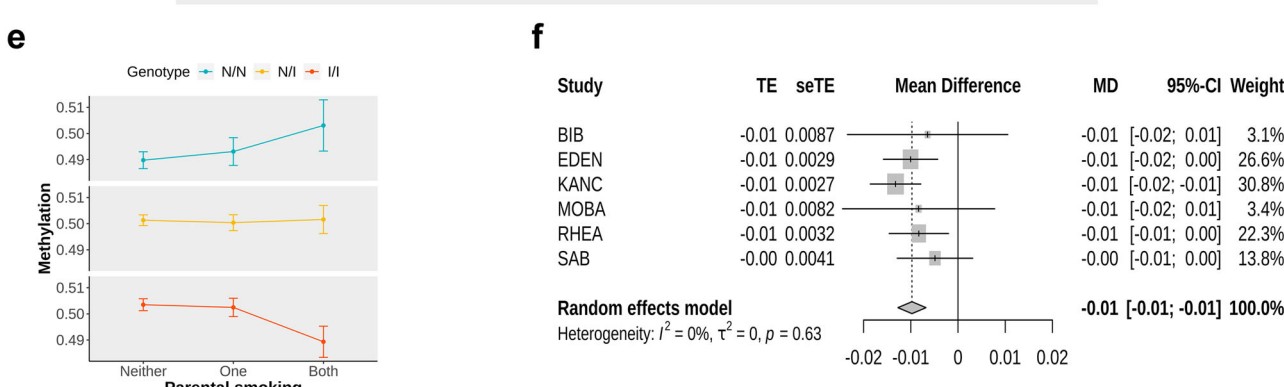

**Fig. 3 Interaction and forest plots for *TDH, GATA4,* and *TRMT9B* genes. a** Interaction plot illustrating differences across inv-8p23.1 genotypes in the association between cg01489256 (*TDH*) methylation and meat intake (expressed in servings per week). Methylation means the given meat-intake status and inversion genotype are represented with their 95% confidence intervals ($N = 1009$). **b** Forest plot showing the meta-analysis effect estimates of inv-8p23.1–meat-intake interaction on cg01489256 methylation across HELIX cohorts. **c** Interaction plot illustrating differences across inv-8p23.1 genotypes in the association between cg26020513 (*GATA4*) methylation and manganese ($N = 1009$). **d** Forest plot showing the meta-analysis effect estimates of inv-8p23.1–manganese interaction on cg26020513 methylation across HELIX cohorts. **e** Interaction plot illustrating differences across inv-8p23.1 genotypes in the association between cg08196601 (*TRMT9B*) methylation and parental smoking ($N = 1009$). **f** Forest plot showing the meta-analysis effect estimates of the inv-8p23.1–parental smoking interaction on cg08196601 methylation across HELIX cohorts. Blue points and lines illustrate noninverted homozygous (N/N), yellow illustrates heterozygous (N/I), and orange illustrates inverted homozygous (I/I) individuals. The error bar represents one standard deviation.

within 5-bp distance and overall population frequency higher than 1%, ruling out technical and genetic variation as main contributors to the methylation differences. We observed that inversions at 8p23.1 and 17q21.31 were strongly characterized by their methylation patterns in the region. However, the effect was less strong for inv-16p11.2, which can be due to the higher number of haplotype groups supported by the inversion, that is, two distinct haplotype groups in the standard allele and one in the inverted allele, and the fact that this inversion is smaller in size (0.45 Mb vs. 0.9 Mb for inv-17q21.31 and almost 4 Mb for inv-8p23.1)[8]. These specific effects on the methylation patterns could be mainly caused by differences in the three-dimensional (3D) DNA configuration for each allele[26], rendering some haplotypes more accessible to the different factors that could facilitate DNA methylation. This mechanism would explain how a recurrent but nonpolymorphic inversion at Xq28 causing Hemophilia A has been associated with specific methylation changes[23] or how de novo inversions at 11p15.5 causing Beckwith–Wiedemann syndrome can be hypermethylated[24]. The possible correlation of inversion haplotypes with different 3D configurations and nuclear localization should be investigated in future studies.

We found that while the effects of the inversion on gene transcription and CpG methylation are widespread across the affected region with some overlap, the specific expression changes driven by inversion-association methylation need to be individually assessed. While the extended pattern of methylation across the inversion can be a consequence of the reconfiguration of the chromatin structure, gene expression may be more susceptible to the tissue and the local genetic variability in linkage with an inversion allele. In the case of 17q21 inversion, for instance, we found clear methylation patterns associated with inversion alleles, but no expression differences, which suggests that these methylation changes would have no relevant consequences in blood. By contrast, we also identified a relevant and specific mediator role by the methylation at promoters of TUFM and SULT1A1 on the associations of their expressions with inv-16p11.2. Remarkably, these are candidate genes in the association between inv-16p11.2 and the co-occurrence of asthma and obesity[8].

Previous studies have reported transcriptomic effects of inv-17q21.31 in blood only in genes with multiple copies[53,54]. This is a complex region with high variability in the gene copies within the inversion alleles, high homology between the genes with multiple copies, and low expression of the genes in blood[14,55]. This could explain the lack of eQTL effects of inv-17q21 in blood that we observed.

We have found that several methylation effects of inversions are modifiable by numerous environmental exposures, suggesting additional inversion-methylation effects to those driven by genetic variability. We observed that inversions significantly interacted with a wide range of exposures affecting DNA methylation across the inverted segments. Therefore, inversions are common copy-neutral polymorphisms that seem to be important contributors to gene-environment interactions, whose detection remains elusive in genomic and high-dimensional exposure data[56–58]. We analyzed data from an exposome study, covering a wide range of exposure families believed to affect children's development. The exposome data included environmental exposures but also exposures from the diet, urban exposome, and chemical compounds[31]. In total, we assessed 64 exposures (7 during pregnancy and 57 at 6–11 years of age) grouped in 12 families. We observed inversion interactions in most of the exposure families, most prominently in metals, diet, phenols, and organochlorines. Validation of these results and their consequences remain to be evaluated. Our results support the notion that inversions can change the way exposures affect a child's development by changing the genetic context. Carriers of genomic variants, such as these inversions that may affect the function of a set of genes in a specific direction, can be more susceptible to (or naturally protected against) disease or developmental disorders if exposed to a relevant environmental risk factor[59]. Thus, allele-specific methylation in response to different environmental factors could also contribute to the positive selection that has been documented for all three inversions in some human populations[8,12,60].

We found numerous significant inversion-exposure interactions on methylation levels in important genes that deserve further study. These include, among others, Alzheimer's MAPT and its associations with copper[61], MSRA's role in repairing oxidative damage to proteins and its relation with diet and parental smoking, and the oncogene WNT3 and its relation to molybdenum and mercury exposure. Here, we highlight three interactions with potential clinical interest and substantial support from previous studies. First, we observed the interaction of inv-8p23.1 with meat intake associated with TDH methylation levels. Remarkably, the inversion, the exposure, and the gene are independently associated with obesity in adults[5,39–41]. Our data revealed that noninverted homozygous individuals, those with a higher risk of obesity, decreased methylation of two CpG sites within TDH as meat intake increases. While further studies are needed to describe the role that this pseudogene plays in obesity during development, it is clear that these need to incorporate the effects of the inversion and its methylation status. In addition, clinical interventions of obesity aiming at managing meat intake should consider the methylation of the gene and the inversion genotype of individuals. Second, we observed that cg26020513 within GATA4 was hypermethylated in blood when manganese exposure increased but only in noninverted homozygous individuals. It is notable that the hypermethylation of cg26020513 has been strongly associated with congenital heart defects in fetuses[42], mutations in GATA4 have been associated with cardiac septal defects[62], and manganese toxicity in heart tissue is well documented[63]. The inversion also interacted with other relevant exposures on GATA4 methylation, including mercury, with reported effects in heart-rate variability in children[64], diethylphosphate, Mediterranean diet, and PCB 138. Therefore, the extent to which the inversion status can protect against the positive association between these exposures and GATA4 methylation deserves further scrutiny. Third, we observed that the effects of tobacco smoke (during pregnancy or in childhood) and air pollution (outdoor PM2.5 exposure) on TRMT9B methylation changed, depending on the inv-8p23.1 genotype. Since these two exposures increase the risk of respiratory diseases[45–47] and TRMT9B is a gene associated with an upper respiratory tract disease[43,44], our results suggest a likely role of the gene in the association between inv-8p23.1 and asthma[5].

To the best of our knowledge, this is the first study to systematically assess the methylation landscape within three common human inversions and its interaction with the exposome. We have shown that genomic inversions are associated with the methylation of the CpG sites within the inversion region and that this association is modulated by a wide range of environmental exposures during childhood.

## Methods

**Study population.** The Human Early Life Exposome (HELIX) project[36] comprises a total of 1301 mother–child pairs from six birth cohorts in Europe: BIB (Born in Bradford; the United Kingdom)[65], EDEN (Etude des Déterminants pré et postnatals du développement et de la santé de l'Enfant; France)[66], INMA-SAB (Infancia y Medio Ambiente; Spain; subcohort Sabadell)[67], KANC (Kaunas cohort; Lithuania)[68], MoBa (The Norwegian Mother, Father and Child Cohort study; Norway)[69], and Rhea (Greece)[70]. These mother–child pairs participated in a common, completely harmonized, follow-up examination between December 2013 and February 2016, when children were between 6 and 11 years old[71]. The main

goal of this project was to implement exposure assessment and biomarker methods to characterize early-life exposure to multiple environmental factors and associate these with omics biomarkers and child health outcomes. For these same children, multi-omics molecular phenotyping was performed, including measurement of blood DNA methylation (450 K, Illumina), blood gene expression (HTA v2.0, Affymetrix), blood miRNA expression (SurePrint Human miRNA rel 21, Agilent), plasma proteins (Luminex), serum metabolites (AbsoluteIDQ p180 kit, Biocrates), urinary metabolites ([1]H NMR spectroscopy), and DNA microarray (Chemagen kit, Perkin Elmer). All studies received approval from the ethics committees of the centers involved and written informed consent was obtained from all participants.

**Molecular phenotypes**

*Inversion genotype data.* DNA was obtained from buffy coat collected in EDTA tubes at 6–11 years of age. Briefly, DNA was extracted using the Chemagen kit (Perkin Elmer) in batches of 12 samples. Samples were extracted by cohort and following their position in the original boxes. DNA concentration was determined in a NanoDrop 1000 UV–Vis Spectrophotometer (ThermoScientific) and with Quant-iT™ PicoGreen® dsDNA Assay Kit (Life Technologies). Genome-wide genotyping was performed using the Infinium Global Screening Array (GSA) MD version 1 (Illumina) at the Human Genomics Facility (HuGe-F), Erasmus MC (www.glimdna.org). Genotype calling was done using the GenTrain2.0 algorithm based on a custom clusterfile for 692,367 variants implemented in the GenomeStudio software. Annotation was done with the GSAMD-24v1-0_20011747_A4 manifest, SNP coordinates were reported on human reference GRCh37 and Source strand (Forward strand report in GenomeStudio). The initial dataset consisted of 1,397 samples and 692,367 variants. Samples with discordant sex, duplicated, contaminated (high heterozygosity), and relatives (IBD > 0.185) were filtered out. SNPs with variant call rate <95%, minimum allele frequency <1%, and HWE *P*-value ($1 \times 10^{-6}$) were excluded. Major population ancestry groups were estimated using Peddy[37] and only individuals of European ancestry were kept in the analysis. The final dataset consisted of 1,009 samples and 509,344 SNP variants. From this dataset, we selected inversions that could be genotyped with *scoreInvHap* and had more than 10 CpG sites in the inversion region: inv-8p23.1, inv-16p11.2, and inv-17q21.31 (Table 2 and Supplementary Tables 2 and 3).

*DNA methylation.* The DNA was obtained using the same methodology as for genetics data. DNA methylation was assessed using the Infinium Human Methylation 450 beadchip (Illumina), following the manufacturer's protocol. *Minfi* R package[72] was used for the preprocessing of DNA methylation data. *MethylAid* package[73] was employed to perform the first quality control of the data. Probes with low call rates were filtered following the guidelines of Lehne et al.[74]. The functional normalization method was further applied, including Noob background subtraction and dye-bias correction[75]. Several quality-control checks were performed: sex consistency using the *shinyMethyl* package[76], consistency of duplicates, and genetic consistency for the samples that had genome-wide genotypic data. Duplicated samples and control samples were removed, as well as probes that measure methylation levels at non-CpG sites[77]. Probes that cross-hybridize were excluded. Moreover, we used InfiniumAnnotation from https://zwdzwd.github.io/InfiniumAnnotation to filter probes where 30-bp 3′-subsequence of the probe is nonunique, probes with INDELs, probes with extension base inconsistent with specified color channel (type I) or CpG (type II) based on mapping, probes with a SNP in the extension base that causes a color-channel switch from the official annotation, and probes where 5-bp 3′-subsequence overlap with any of the SNPs with global population frequency higher than 1%. Consequently, the number of CpG probes analyzed was 371,533, initially available for 1192 subjects. We then used Combat algorithm to remove the batch effects supported by the slide. Methylation levels were expressed as beta values (average methylation levels for an individual, between 0 for a never-methylated CpG site and 1 for an always-methylated CpG site) and CpG sites were annotated to genes by Illumina HM450 manifest file (version 1.2). We discarded the subjects without inversion-status data and without European ancestry based on genomic data, resulting in 1009 individuals for the analysis. For each inversion, we selected the CpG sites contained in the inversion region ±1 Mb, resulting in 848 CpG sites for inv-8p23.1, 401 for inv-16p11.2, and 666 for inv-17q21.31 (Table 2 and Supplementary Table 2). Blood cell-type proportions were estimated from methylation data according to Houseman et al. algorithm[78] and Reinius reference panel[79].

*Gene expression.* At the period of clinical examination that took place when children were between 6 and 11 years old, RNA was extracted from whole blood collected in Tempus tubes. Samples with RIN > 5 were considered. Gene expression was assessed using the GeneChip® Human Transcriptome Array 2.0 (HTA 2.0) (Affymetrix, USA) at the University of Santiago de Compostela (USC, Spain), following the manufacturer's protocol. Samples were randomized and balanced by sex and cohort within each batch. Data were normalized at the gene level with the GCCN (SST-RMA) algorithm, and batch effects and blood cell-type composition were controlled with two surrogate variable analysis (SVA) methods, *isva*[80] and *SmartSVA*[81], during the differential expression analyses. Gene expression values were log2 transformed, and annotation of transcript clusters (TCs) to genes was done with NetAffx annotation (version 36). Genes without Gene Symbol annotation or with call rate <20% were removed, restricting to 25,255 genes. From this number of genes, we selected those within the inversion regions ±1 Mb (inv-8p23.1: 83 genes; inv-16p11.2: 58 genes; inv-17q21.31: 61 genes). From a total of 1158 subjects that had transcriptomic data, we selected individuals with European ancestry (based on genomic data) who had available inversion-status data and cell-type proportions assessed from methylation data, resulting in a total of 790 subjects (Table 2 and Supplementary Table 1).

**Exposome assessment**. The assessment of the exposome has been previously published[82]. In our study, we included 7 exposures assessed during pregnancy and 57 exposures assessed during childhood at age 6–11 y (Supplementary Data 4). These 64 exposures were selected from the entire exposome dataset according to the number of missing values they had. We did not include exposures that had more than 10% of missings in the whole dataset or with more than 20% missing in one or more cohorts. We also excluded exposures whose levels were not present in all cohorts. Third, we selected the most representative exposures within each family.

The pregnancy exposome consists of 7 exposures, including outdoor PM2.5, normalized difference vegetation index (NDVI), 4 PFASs, and maternal smoking during pregnancy. The postnatal exposome was divided into 12 exposure families: outdoor air pollution (2), building environment (1), diet (6), metals (9), natural spaces (1), organochlorines—OCs (8), organophosphate pesticides—OP pesticides (5), polybrominated diphenyl ethers—PBDEs (2), perfluorinated alkylated substances—PFAS (5), phenols (7), phthalates (10), and second-hand exposure to tobacco smoke (1) (Fig. 2a). Metals, OCs, OP pesticides, PBDEs, PFASs, phenols, and phthalates were assessed by biomarkers in children at the time of the clinical examination, from a pool of two urine samples or one serum sample[83]. Air pollution, natural spaces, and building environment quantification were assessed during the year before child examination or during pregnancy by environmental geographic information systems (GIS). Tobacco smoke and diet were evaluated by questionnaires. Missing values for all exposures were imputed using the method of chained equations[84], as described in detail elsewhere[82]. Most exposure variables were transformed as described in Supplementary Data 4.

**Fetal heart-tissue samples**. Human fetal samples from 40 fetuses of terminated pregnancies due to a major congenital heart defect (gestational age 21–22 weeks in all cases) were obtained from Biobanc Hospital Universitari Vall d'Hebron (HUVH) in a related project addressed to define the genetic and epigenetic basis of congenital heart defects[38]. Informed consent was obtained from parents and the study was approved by the institutional ethics committee. Heart-tissue DNA was obtained following necropsy using standard procedures, whole-genome sequencing was performed at Centogene, and DNA methylation was measured with Infinium MethylationEPIC[38].

After quality control, one sample was discarded (Supplementary Table 4). During the preprocessing of methylation data, probes with a single-nucleotide polymorphism (SNP) with overall population frequency higher than 1% based on InfiniumAnnotation from https://zwdzwd.github.io/InfiniumAnnotation were removed. Selecting the CpG sites within the inversion region ±1 Mb, we analyzed 898 CpG sites from inv-8p23.1, 409 from inv-16p11.2, and 698 from inv-17q21.31.

**Statistics and reproducibility**

*Genome-wide analysis.* Differential methylation analyses were performed using *MEAL* Bioconductor's package[85]. We performed a differential mean analysis

---

**Table 2 Characteristics of HELIX data relating 3 common polymorphic inversions in humans.**

| Genomic inversion | Length (kb) | Inversion region ±1 Mb | Inversion frequency (%) | Omics | Number of samples | Number of features |
|---|---|---|---|---|---|---|
| 8p23.1 | 3924.86 | chr8:7055789-12980649 | 57.95 | Methylome | 1009 | 848 |
| | | | | Transcriptome | 926 | 83 |
| 16p11.2 | 364.17 | chr16:27424774-29788943 | 34.49 | Methylome | 1009 | 401 |
| | | | | Transcriptome | 926 | 58 |
| 17q21.31 | 710.89 | chr17:42661775-45372665 | 23.96 | Methylome | 1009 | 666 |
| | | | | Transcriptome | 926 | 61 |

The table shows the length in kb, the mapping coordinates hg19 ±1 Mb, the frequency of all the inversions obtained from *scoreInvHap*[11], and the number of samples and features used in transcriptome and methylome analysis for each inversion.

(DMA) on inversion genotypes using the function *runDiffMeanAnalysis* that calls *limma*[86]. Based on a priori knowledge, we adjusted all the regression models by sex, age, population stratification (using the first 10 principal components of the GWAS that highly correlated with cohort), and cell type (Supplementary Tables 1 and 2). To correct for the variance between cohorts, we performed this analysis for each cohort separately, and we meta-analyzed the results using the function *metagen* from *meta* package[87]. For each inversion, in each cohort, we fitted models

$$E_j = \alpha_j + \beta_{jk} I_k + \Sigma_r \gamma_r C_r + \varepsilon_j \quad (1)$$

where $E_j$ is the methylation or expression-level vector across individuals at probe $j$, $I_k$ are the individuals' genotypes for inversion $k$ (8p23.1, 16p11.2, and 17q21.31), $C_r$ is the $r$ covariate and its effect $\gamma_r$, and $\varepsilon_j$ is the noise that follows the distribution of methylation or expression levels with mean 0. $\beta_{jk}$ is the effect of interest measuring the effect of the inversion. The $\beta_{jk}$ were then meta-analyzed across cohorts. P-values derived from the meta-analyses were corrected for multiple comparisons for the number of probes using Bonferroni's correction. The inflation or deflation of P-values across the methylome or transcriptome was tested with Q–Q plots.

*Exposome-wide interaction analysis*. Based on the genome-wide analysis, the same functions were implemented for the exposome-wide interaction analysis. In this case, the effect of interest was the inversion-exposure interaction in the model

$$E_j = \alpha_j + \beta_{jik}(X_i \times I_k) + \Sigma_r \gamma_r C_r + \varepsilon_j \quad (2)$$

where $X_i$ is the level of exposure $i$ across individuals. $\beta_{jik}$ is the effect of interest given by the exposure-inversion interaction. In this case, the covariates also included exposure $i$, the inversion genotypes, maternal education level, and child body mass index (BMI). P-values were corrected for multiple comparisons across CpG sites and exposures using Bonferroni's correction. The inflation or deflation of P-values across the methylome was tested with Q–Q plots.

**Reporting summary**. Further information on research design is available in the Nature Research Reporting Summary linked to this article.

## Data availability

Source data underlying Figs. 2a, 3a and e are available in Supplementary Data 7. The HELIX data warehouse has been established as an accessible resource for collaborative research involving researchers external to the project. Access to HELIX data is based on approval by the HELIX Project Executive Committee and by the individual cohorts. Further details on the content of the data warehouse (data catalog) and procedures for external access are described on the project website (http://www.projecthelix.eu/index.php/es/data-inventory). The data used in this analysis are not available for replication because specific approvals from HELIX Project Executive Committee and the University of Southern California Institutional Review Board must be obtained to access them. Please contact the corresponding author for more information regarding access to HELIX data.

## Code availability

Any custom code or software used in our analysis is available at https://doi.org/10.5281/zenodo.6417926 (URL: https://zenodo.org/badge/latestdoi/296552532).

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

## Acknowledgements

We are grateful to all the participating children, parents, practitioners, and researchers in the six countries who took part in this study. The study has received funding from the European Community's Seventh Framework Programme (FP7/2007–2013) under grant agreement no 308333 (HELIX project), and the H2020-EU.3.1.2.—Preventing Disease Programme under grant agreement no 874583 (ATHLETE project). The HELIX genotyping was supported by the projects PI17/01225 and PI17/01935, funded by the Instituto de Salud Carlos III and cofunded by European Union (ERDF, "A way to make Europe") and the Centro Nacional de Genotipado-CEGEN (PRB2-ISCIII). BiB received core infrastructure funding from the Wellcome Trust (WT101597MA) and a joint grant from the UK Medical Research Council (MRC) and Economic and Social Science Research Council (ESRC) (MR/N024397/1). INMA-SAB data collections were supported by grants from the Instituto de Salud Carlos III, CIBERESP, and the Generalitat de Catalunya-CIRIT. KANC was funded by the grant of the Lithuanian Agency for Science Innovation and Technology (6-04-2014_31V-66). The Norwegian Mother, Father and Child Cohort Study is supported by the Norwegian Ministry of Health and Care Services and the Ministry of Education and Research. The Rhea project was financially supported by European projects (EU FP6-2003-Food-3-NewGeneris, EU FP6. STREP Hiwate, EU FP7 ENV.2007.1.2.2.2. Project No 211250 Escape, EU FP7-2008-ENV-1.2.1.4 Envirogenomarkers, EU FP7-HEALTH-2009- single stage CHICOS, EU FP7 ENV.2008.1.2.1.6. Proposal No 226285 ENRIECO, EU FP7-HEALTH-2012 Proposal No 308333 HELIX), and the Greek Ministry of Health (Program of Prevention of obesity and neurodevelopmental disorders in preschool children, in Heraklion district, Crete, Greece: 2011–2014; "Rhea Plus": Primary Prevention Program of Environmental Risk Factors for Reproductive Health, and Child Health: 2012–15). This research has received funding from the Spanish Ministry of Education, Innovation and Universities, the National Agency for Research and the Fund for Regional Development (RTI2018-100789-B-I00), MaratóTV3 (2015–3230), the Spanish Ministry of Science and Innovation through the "Centro de Excelencia Severo Ochoa 2019-2023 (CEX2018-000806-S) and Maria de Maeztu (MDM-2014-0370)" Programs, and support from the Generalitat de Catalunya through the CERCA and Consolidated Research Group (2017SGR01974) Programs. NC and JU are supported by Spanish regional program PERIS (Ref.: SLT017/20/000061 and SLT017/20/000119, respectively), granted by Departament de Salut de la Generalitat de Catalunya. We thank Pau Bosch Castro for designing and creating the featured image.

## Author contributions

J.R.G. conceived the study and supervised genomic inversion analyses. J.R.G., A.C., and L.A.P.-J. designed the analysis. L.B.-D. performed genomic inversion calling and N.C.-G. the statistical analyses. M.V. coordinates the HELIX project, J.U. is the data manager, and L.M. is the scientific coordinator. M.B., J.W., R.S., M.C., J.R.G., and M.V. designed the omics study in HELIX. The following authors participated in omics data acquisition and quality control: G.E. (genomics), M.B. (transcriptomics and DNA methylation), A.C. (DNA methylation), M.J.N. (exposome), and C.T. (exposome). J.W., R.G., M.V., R.S., L.C., and C.T. are the PIs of the cohorts. T.Y., S.A., M.C., J.L., N.S., and K.G. participated in sample and data acquisition. C.R.-A. performed inversion-methylation analyses in heart tissue. L.A.P.-J. coordinated the study of heart tissue in CHD. N.C.-G. and A.C. cowrote the original draft of the paper and J.R.G., L.A.P.-J., M.B., C.R.-A., and L.B.-D. contributed to review and edit the paper. All authors read and approved the final version of the paper.

## Competing interests

The authors declare no competing interests.
