## [Peer Review File · Communications Biology]

Reviewers' comments:

Reviewer #1 (Remarks to the Author):

Review COMMSBIO-21-2898-T

The effect of polymorphic inversions on DNA methylation and its modulation by the early life exposome

Summary

The manuscript describes DNA methylation patterns of three common human inversions, 8p23.1, 16p11.2, and 17q21.31. Two sample sets were investigated, the HELIX cohort (n=1301, blood methylation and transcriptome) and samples from interrupted pregnancies due to congenital heart defects (n=39, fetal heart methylation). For both datasets, inversion-specific methylation patterns were detected. Furthermore, numerous significant exposures were identified that impacted the inversion-methylation levels.

Major comments:

- The method used to detect the inversions, searching for specific haplotypes, could be interesting to use also in other genomics projects. Inversion variants are often overlooked since specific callers and methods are necessary to detect them.
- The methylation data is more difficult to interpret. Is the observed pattern due to the inversion or due to the specific haplotype. In addition to the inversion the carriers will also share 100-1000 SNVs. Those SNVs are in fact used to detect the inversion and the question is what is causing the methylation pattern. The inversion or the other thousands of variants. There could also be CNVs present in the inverted alleles and as far as I can tell this was not investigated.
- If, the authors want to claim that the inversions cause the aberrant methylation patterns they should investigate rare inversions as well and see if they can detect similar alternative methylation compared to controls. Alternatively, they could pick specific haplotypes and compare inversion carriers vs non-carriers and check if the differences remain.

Minor comments

- Subsection header missing for introduction

Reviewer #2 (Remarks to the Author):

The authors find that the alleles of 3 different inversions are associated with distinctive methylation patterns in CpG dinucleotides. These methylation patterns can be modified by exposures to several environmental or diet elements, with different methylation changes taking place in different inversion alleles in some cases, revealing an interaction between inversions, methylation patterns, and exposures. This article explores the effects of inversion alleles in DNA methylation, which is a potential effect of inversions that has not been analyzed before in detail. The results are interesting and that is why I recommend it for publication with minor changes.

The analyses performed in this work are very clearly explained, with complete and detailed methods. My main concern is related to the relevance of these methylation changes associated with inversion alleles. The authors emphasize in the discussion the importance of methylation in mediating the phenotypical effects of inversions. However, they do not provide evidence of the role of these methylation changes observed between inversion alleles in the final gene expression levels. For example, they give the number of elements that are hypo or hyper methylated in each inversion, and the genes that are differentially expressed in each inversion, but they do not explain how these two categories fit with each other, that is, if these methylation changes are located in the differentially expressed genes, or if the observed expression changes are in the expected directions based on the methylation of these regions (given that usually DNA methylation involves silencing).

If the differentially-expressed genes do not correspond to those differentially methylated, or if expression differences do not fit the hypo/hypermethylation patterns observed in a given inversion allele, it might indicate that there are other factors that might explain better (or contribute more to) the observed expression differences than methylation changes (like the different haplotypes in inverted and non-inverted chromosomes, or the alteration of regulatory elements by the position change of genes). In the case of 17q21 inversion, there are different methylation patterns associated with inversion alleles, but no expression differences, which suggests that these methylation changes would have no relevant consequences in blood. Is it possible to determine the relevance of these methylation changes? In how many genes might methylation contribute significantly to change gene expression and phenotype? In case that there are other factors that contribute more importantly to gene expression differences, maybe the modifier or complementary role of methylation should be acknowledged in the discussion.

Following the same idea, and taking into account the exposome, are there expression changes in the corresponding gene in those individuals that present a different methylation due to the interaction of inversion and exposure? For example, is TDH differentially expressed between NI/NI and I/I with different methylation in those individuals exposed to high meat consumption? If there is no expression change, which are then the consequences of this differential methylation? How could it affect phenotype?

Other points that I would like authors to clarify are:

Why are only 3 inversions analyzed? Based on previous publications, the authors have data on more human polymorphic inversions to which this analysis could be extended. Which are the criteria to choose only these 3?

I would like the authors to comment if the observed methylation changes associated with inversion alleles or with a given exposure are the same in different tissues. For example, are the same positions differentially methylated in blood and fetal heart with respect to inversion alleles? Or are inversion alleles associated with different methylation patterns in different tissues? Also, the authors analyze blood samples, but probably some of the genes in the analyzed regions perform their main roles in other tissues where methylation changes could have more visible and relevant consequences than in blood. How many genes of those analyzed in these regions are actually expressed in blood, so that the main effects of methylation changes would be expected in the analyzed tissue?

The authors claim that inversion 17q21 is not an eQTL for any gene in blood cells. However, two recent works that analyzed this well-known inversion show that it affects the expression of many genes in the region, some of them in several tissues, including blood (see Puig et al. 2020 Genome Research, or Degenhardt et al. 2021 medRxiv). I would like the authors to discuss this discrepancy.

Finally, some minor changes:

There is a number 4 next to Carlos Ruiz-Arenas name in the author list.

The official gene symbols according to HGNC nomenclature (<https://www.genenames.org/>) for KIAA1267 and c8orf79 are KANSL1 and TRMT9B, respectively.

The exposure to "outdoor PM2.5" appears mentioned in page 8 without any definition, and it is not until page 11 when it is explained as "air pollution". This explanation should be moved to the first mention of the exposure in page 8.

Figure 3 legend: "meat intake" instead of "meet intake".

Reviewer #3 (Remarks to the Author):

The effect of polymorphic inversions on DNA methylation and its modulation by the early-life exposome

Carreras-Gallo et al. 2021

The authors find that the alleles of 3 different inversions are associated with distinctive methylation patterns in CpG dinucleotides. These methylation patterns can be modified by exposures to several environmental or diet elements, with different methylation changes taking place in different inversion alleles in some cases, revealing an interaction between inversions, methylation patterns, and exposures. This article explores the effects of inversion alleles in DNA methylation, which is a potential effect of inversions that has not been analyzed before in detail. The results are interesting and that is why I recommend it for publication with minor changes.

The analyses performed in this work are very clearly explained, with complete and detailed methods. My main concern is related to the relevance of these methylation changes associated with inversion alleles. The authors emphasize in the discussion the importance of methylation in mediating the phenotypical effects of inversions. However, they do not provide evidence of the role of these methylation changes observed between inversion alleles in the final gene expression levels. For example, they give the number of elements that are hypo or hyper methylated in each inversion, and the genes that are differentially expressed in each inversion, but they do not explain how these two categories fit with each other, that is, if these methylation changes are located in the differentially expressed genes, or if the observed expression changes are in the expected directions based on the methylation of these regions (given that usually DNA methylation involves silencing).

If the differentially-expressed genes do not correspond to those differentially methylated, or if expression differences do not fit the hypo/hypermethylation patterns observed in a given inversion allele, it might indicate that there are other factors that might explain better (or contribute more to) the observed expression differences than methylation changes (like the different haplotypes in inverted and non-inverted chromosomes, or the alteration of regulatory elements by the position change of genes). In the case of 17q21 inversion, there are different methylation patterns associated with inversion alleles, but no expression differences, which suggests that these methylation changes would have no relevant consequences in blood. Is it possible to determine the relevance of these methylation changes? In how many genes might methylation contribute significantly to change gene expression and phenotype? In case that there are other factors that contribute more importantly to gene expression differences, maybe the modifier or complementary role of methylation should be acknowledged in the discussion.

Following the same idea, and taking into account the exposome, are there expression changes in the corresponding gene in those individuals that present a different methylation due to the interaction of inversion and exposure? For example, is TDH differentially expressed between NI/NI and I/I with different methylation in those individuals exposed to high meat consumption? If there is no expression change, which are then the consequences of this differential methylation? How could it affect phenotype?

Other points that I would like authors to clarify are:

Why are only 3 inversions analyzed? Based on previous publications, the authors have data on more human polymorphic inversions to which this analysis could be extended. Which are the criteria to choose only these 3?

I would like the authors to comment if the observed methylation changes associated with inversion alleles or with a given exposure are the same in different tissues. For example, are the same positions differentially methylated in blood and fetal heart with respect to inversion alleles? Or are inversion alleles associated with different methylation patterns in different tissues? Also, the authors analyze blood samples, but probably some of the genes in the analyzed regions perform their main roles in other tissues where methylation changes could have more visible and relevant consequences than in blood. How many genes of those analyzed in these regions are actually expressed in blood, so that the main effects of methylation changes would be expected in the

analyzed tissue?

The authors claim that inversion 17q21 is not an eQTL for any gene in blood cells. However, two recent works that analyzed this well-known inversion show that it affects the expression of many genes in the region, some of them in several tissues, including blood (see Puig et al. 2020 Genome Research, or Degenhardt et al. 2021 medRxiv). I would like the authors to discuss this discrepancy.

Finally, some minor changes:

There is a number 4 next to Carlos Ruiz-Arenas name in the author list.

The official gene symbols according to HGNC nomenclature (<https://www.genenames.org/>) for KIAA1267 and c8orf79 are KANSL1 and TRMT9B, respectively.

The exposure to "outdoor PM2.5" appears mentioned in page 8 without any definition, and it is not until page 11 when it is explained as "air pollution". This explanation should be moved to the first mention of the exposure in page 8.

Figure 3 legend: "meat intake" instead of "meet intake".

Reviewer #1: The manuscript describes DNA methylation patterns of three common human inversions, 8p23.1, 16p11.2, and 17q21.31. Two sample sets were investigated, the HELIX cohort (n=1301, blood methylation and transcriptome) and samples from interrupted pregnancies due to congenital heart defects (n=39, fetal heart methylation). For both datasets, inversion-specific methylation patterns were detected. Furthermore, numerous significant exposures were identified that impacted the inversion-methylation levels.

We thank the reviewer for the comments, in particular, those relating to the causal relationship between the methylation patterns and the inversion. The comments have allowed us to give a better context to our findings, discuss their correct interpretation and highlight their significance. As a result, we have included a new paragraph in the discussion and a justification for the selection of the inversions that we study in methods.

Major concerns:

(1) The method used to detect the inversions, searching for specific haplotypes, could be interesting to use also in other genomics projects. Inversion variants are often overlooked since specific callers and methods are necessary to detect them.

Response: We really appreciate the positive comment of the reviewer about the method designed by our group used to genotype inversions. Indeed, inversions are structural variants with relevant effects on common diseases, and, using SNPs, they can be genotyped easily employing *scoreInvHap*.

(2) The methylation data is more difficult to interpret. Is the observed pattern due to the inversion or due to the specific haplotype? In addition to the inversion the carriers will also share 100-1000 SNVs. Those SNVs are in fact used to detect the inversion and the question is what is causing the methylation pattern. The inversion or the other thousands of variants. There could also be CNVs present in the inverted alleles and as far as I can tell this was not investigated.

Response: We agree that elucidating the causes of specific CpG associations with the inversion is difficult and needs special attention in the discussion. It is challenging to know whether a CpG association is due to the inverted sequence itself or to an unobserved variant locked in an inversion state. However, we would like to highlight two points. First, in relation to SNVs, we accounted for CpG associations due to SNVs, removing all the CpG sites with a SNV within 5bp distance, as mentioned in the manuscript (page 18, line 369):

“We removed probes with SNPs within 5bp distance and overall population frequency higher than 1%, ruling out technical and genetic variation as main contributors to the methylation differences.”

Second, our observations reveal a spatial pattern given by the correlation of several CpG associations that fits the extension of the inversion. While individual CpG associations may be primarily due to unobserved variants, it is clear that the cause of such extended pattern along the affected sequence has been produced by the presence of the inversion. When we performed

A partnership of:

the principal component (PC) analysis of all the CpG sites from each inversion region, we were able to see differences between the inversion genotypes. As the PCs summarize the methylation variability between individuals across the entire region, we can assume that the methylation patterns underlying the PC clustering are caused by the inversion itself, likely due to both the DNA reconfiguration and the accumulation of specific genetic variability along the segment that results from the suppression of recombination between inversion states.

We have added a sentence in the discussion to address this issue (page 17, line 345):

“While individual CpG associations with the inversion may be due to the inversion or to local genetic variability in linkage with the inversion, our observations in the PC analysis reveal a spatial pattern given by the correlation of several CpG sites associations that fits the extension of the inversion. It is clear that the cause of such extended pattern along the affected sequence has been produced by the presence of the inversion, likely due to both the DNA reconfiguration and the accumulation of specific genetic variability along the segment that results from the suppression of recombination between inversion states.”

(3) If, the authors want to claim that the inversions cause the aberrant methylation patterns they should investigate rare inversions as well and see if they can detect similar alternative methylation compared to controls. Alternatively, they could pick specific haplotypes and compare inversion carrier’s vs non-carrier’s and check if the differences remain.

Response: Referring to our answer to the previous comment, we claim that the inversion underlies the extended spatial pattern given by the correlation among several CpG associations that fits the extension of the inversion. As such, it is difficult to see how a local variant or haplotype can explain the extension of the patterns.

We agree that the study of rare inversions is an interesting approach to determine the CpG methylation associated with the sequence rearrangement of an inversion. Although there are no studies evaluating the inversion region of rare inversions as a whole, there are studies demonstrating the methylation changes in regions enriched of CpG sites in diseases caused by an inversion (Jamil et al. 2019, Front. Genet). Our study, however, is based on common, historic, non-recurrent inversions because they are the ones that can be reliably genotyped with SNP data. Rare inversions are out of the reach of work. Furthermore, we have limited our study to three large common inversions because other polymorphic inversions that can be genotyped with *scoreInvHap* do not contain CpG sites due to their small size. We have included the following sentence in the methodology (page 21, line 477) and the Supplementary Table S9 with the number of CpG sites located in each inversion:

“From this dataset, we selected inversions that could be genotyped with *scoreInvHap* and had more than 10 CpG sites in the inversion region: inv-8p23.1, inv-16p11.2, and inv-17q21.31 (Table 2 and Table S8-9).”

Minor concerns:

(1) Subsection header missing for introduction.

Response: We did not include an introduction header following the journal guidelines.

Reviewers #2-3: The authors find that the alleles of 3 different inversions are associated with distinctive methylation patterns in CpG dinucleotides. These methylation patterns can be modified by exposures to several environmental or diet elements, with different methylation changes taking place in different inversion alleles in some cases, revealing an interaction between inversions, methylation patterns, and exposures. This article explores the effects of inversion alleles in DNA methylation, which is a potential effect of inversions that has not been analyzed before in detail. The results are interesting and that is why I recommend it for publication with minor changes.

We thank the reviewers for their positive comments on the manuscript. The reviewers' comments have encouraged us to look more in detail at the extent to which the changes of methylation with the inversion are reflected in gene expression changes. As a result, we have performed new comparisons and analyses, and have included new results. We believe that, by addressing their comments, we have increased the clarity and reach of our work.

Major concerns:

(1) The analyses performed in this work are very clearly explained, with complete and detailed methods. My main concern is related to the relevance of these methylation changes associated with inversion alleles. The authors emphasize in the discussion the importance of methylation in mediating the phenotypical effects of inversions. However, they do not provide evidence of the role of these methylation changes observed between inversion alleles in the final gene expression levels. For example, they give the number of elements that are hypo or hyper methylated in each inversion, and the genes that are differentially expressed in each inversion, but they do not explain how these two categories fit with each other, that is, if these methylation changes are located in the differentially expressed genes, or if the observed expression changes are in the expected directions based on the methylation of these regions (given that usually DNA methylation involves silencing).

Response: We thank the reviewer for encouraging us to clarify this point and go further in this direction. We have made further analyses and rewritten parts of the results having the reviewer's comment in mind. In particular, we now explicitly describe the methylation changes that locate in differentially expressed genes and observed specific cases where the changes of the expression and methylation with the inverted allele are consistent. We also performed mediation analysis on specific findings. This new addition replaces our previous paragraph on cis-eQTMs, which, we realized, was not the most convenient comparison. In the results section, we have added the following text (page 9, line 183) with the new Supplementary Figure S4:

"To establish the degree to which the effect of the inversion status in CpG methylation is associated with changes in gene expression of surrounding genes, we searched for the methylation changes that locate in differentially expressed genes (Figure S4). We observed that four genes (*BLK*, *FDFT1*, *XKR6*, and *FAM167A*) overlapped for the inv-8p23.1 with differentially methylated CpG sites. We analyzed whether the observed expression changes were in the expected directions based on the methylation of these regions; that is, hypermethylation of the promoters for downregulated genes, hypomethylation of the promoters for upregulated genes, and hypermethylation of the bodies for upregulated genes. *XKR6* was a highly consistent case whose downregulation and methylation, across 11 CpG sites within its body, were associated

with the inverted allele. For inv-16p11.2, we observed four genes that were differentially expressed and methylated by the inversion allele (*TUFM*, *SBK1*, *SPNS1*, and *SULT1A1*). In this case, most of the CpG sites were in the promoter region (TSS1500) and the relation between the expression and methylation levels was consistent. We further observed that *SULT1A1* and *TUFM* had CpG sites in their promoters (cg01378222 and cg00348858) that highly associated with the effect of inversion in gene expression. We found that cg01378222 mediated the 95% of the association between inv-16p11.2 and the expression of *SULT1A1* ($P < 2 \times 10^{-16}$), and that cg00348858 mediated the 5% of the association between the inversion and *TUFM* expression ($P = 0.002$).

Figure S4 | Venn diagram comparing the genes differentially expressed (eQTL) and methylated (mQTL) by the inversion. a) Inv-8p23.1; b) Inv-16p11.2; c) Inv-17q21.31. The genes that overlapped are annotated with their gene symbol.

We also included a column (“Group Gene”) specifying the position of the CpG site within the gene in the Supplementary Tables S2 and S5.

(2) If the differentially-expressed genes do not correspond to those differentially methylated, or if expression differences do not fit the hypo/hypermethylation patterns observed in a given inversion allele, it might indicate that there are other factors that might explain better (or contribute more to) the observed expression differences than methylation changes (like the different haplotypes in inverted and non-inverted chromosomes, or the alteration of regulatory elements by the position change of genes). In the case of 17q21 inversion, there are different methylation patterns associated with inversion alleles, but no expression differences, which suggests that these methylation changes would have no relevant consequences in blood. Is it possible to determine the relevance of these methylation changes? In how many genes might methylation contribute significantly to change gene expression and phenotype? In case that there are other factors that contribute more importantly to gene expression differences, maybe the modifier or complementary role of methylation should be acknowledged in the discussion.

Response: We agree that the discrepancy between expression and methylation changes needs to be further discussed, particularly in relation to the new results discussed in the previous answer. Two important factors need to be acknowledged, such as the relevance of the tissue and the likely larger contribution of the genetic variability associated with the given inversion allele. We have included the following paragraph in the discussion (page 18, line 383):

“We found that while the effects of the inversion on gene transcription and CpG methylation are widespread across the affected region with some overlap, the specific expression changes driven by inversion-association methylation need to be individually assessed. While the extended pattern of methylation across the inversion can be a consequence of the reconfiguration of the chromatin structure, gene expression may be more susceptible to the tissue and the local genetic variability in linkage with an inversion allele. In the case of 17q21 inversion, for instance, we found clear methylation patterns associated with inversion alleles, but no expression differences, which suggests that these methylation changes would have no relevant consequences in blood. By contrast, we also identified a relevant and specific mediator role by the methylation at promoters of *TUFM* and *SULT1A1* on the associations of their expressions with inv16p11.2. Remarkably, these are candidate genes in the association between inv-16p11.2 and the co-occurrence of asthma and obesity⁸.”

(3) Following the same idea, and taking into account the exposome, are there expression changes in the corresponding gene in those individuals that present a different methylation due to the interaction of inversion and exposure? For example, is *TDH* differentially expressed between NI/NI and I/I with different methylation in those individuals exposed to high meat consumption? If there is no expression change, which are then the consequences of this differential methylation? How could it affect phenotype?

Response: We thank again the reviewers for encouraging us to go further into the interpretation of our results. To answer this question, we have tested the effect of the interaction exposure-inversion on gene expression for our main findings described in Figure 3. For *TDH*, we found a negative correlation between the meat intake and the gene expression ($P = 0.00398$) and a significant association of gene expression with the CpG-inversion interaction, adjusting by sex, age, and cohort ($P = 0.00193$) (New Supplementary Figure S5). Therefore, the increase of meat intake reduces the expression of *TDH* gene in all the individuals while the associated methylation effect depends on the genetic context given by the inversion.

We have added the following text (page 15, line 294) and the Figure S5 in “Genes with strongest and most numerous inversion-exposure interactions” section:

“We further observed that the increase of meat intake reduced the expression of *TDH* ($P = 0.00398$) while the associated methylation effect on the expression depended on the genetic context given by the inversion, adjusting by sex, age, and cohort (CpG-inversion interaction, $P = 0.00193$) (Figure S5).”

Figure S5 | Relationship between the expression and the CpG methylation of TDH gene stratified by inv-8p23.1 genotype. Blue points and lines illustrate non-inverted homozygous (N/N), yellow illustrate heterozygous (N/I), and orange illustrate inverted homozygous (I/I) individuals for inv-8p23.1.

(4) Other points that I would like authors to clarify are: Why are only 3 inversions analyzed? Based on previous publications, the authors have data on more human polymorphic inversions to which this analysis could be extended. Which are the criteria to choose only these 3?

Response: We thank the reviewers for pointing out this issue that needs further clarification. We have included the following sentence in the methodology (page 21, line 477) and the Supplementary Table S9 with the number of CpG sites located in each inversion:

“From this dataset, we selected inversions that could be genotyped with scoreInvHap and had more than 10 CpG sites in the inversion region: inv-8p23.1, inv-16p11.2, and inv-17q21.31 (Table 2 and Table S8-9).”

(5) I would like the authors to comment if the observed methylation changes associated with inversion alleles or with a given exposure are the same in different tissues. For example, are the same positions differentially methylated in blood and fetal heart with respect to inversion alleles? Or are inversion alleles associated with different methylation patterns in different tissues?

Response: This is an interesting point. While we cannot respond fully to this question, as we would have to analyze data from different tissues, we compared our results between blood and heart tissue and observed an important overlap that we now describe in results (page 10, line 227):

“Additionally, we saw that 38 CpG significant sites overlapped between heart (nominal P -value) and blood (adjusted P -value) tissues, 32 of which were in the same direction, suggesting that the effect of inversions on CpG methylation may be sustained between tissues and stages of life.”

(6) Also, the authors analyze blood samples, but probably some of the genes in the analyzed regions perform their main roles in other tissues where methylation changes could have more visible and relevant consequences than in blood. How many genes of those analyzed in these regions are actually expressed in blood, so that the main effects of methylation changes would be expected in the analyzed tissue?

Response: We decided to filter by call rate 20%. Although this does not mean whether the genes are functional or not, it indicates that the genes with expressions in the background noise are removed. Therefore, we are selecting genes that are expressed, at least at low frequencies.

(7) The authors claim that inversion 17q21 is not an eQTL for any gene in blood cells. However, two recent works that analyzed this well-known inversion show that it affects the expression of many genes in the region, some of them in several tissues, including blood (see Puig et al. 2020 Genome Research, or Degenhardt et al. 2021 medRxiv). I would like the authors to discuss this discrepancy.

Response: One possible explanation is that these studies report associations in genes with multiple copies, as they are in the segmental duplications and for which accurate annotation may not be available. We have discussed this discrepancy in the discussion section (page 19, line 393) as follows:

“Previous studies have reported transcriptomic effects of inv-17q21.31 in blood only in genes with multiple copies^{53,54}. This is a complex region with high variability in the gene copies within the inversion alleles, high homology between the genes with multiple copies, and low expression of the genes in blood^{14,55}. This could explain the lack of eQTL effects of inv-17q21 in blood that we observed.”

Minor concerns:

(1) There is a number 4 next to Carlos Ruiz-Arenas name in the author list.

Response: Corrected.

(2) The official gene symbols according to HGNC nomenclature (<https://www.genenames.org/>) for KIAA1267 and c8orf79 are KANSL1 and TRMT9B, respectively.

Response: We completely agree with the reviewer and changed these gene symbols.

(3) The exposure to “outdoor PM2.5” appears mentioned in page 8 without any definition, and it is not until page 11 when it is explained as “air pollution”. This explanation should be moved to the first mention of the exposure in page 8.

Response: We thank the reviewer for the comment on the missing information. Now we include the explanation of the exposure when we first mention it (page 15, line 317).

(4) Figure 3 legend: “meat intake” instead of “meet intake”.

Response: Corrected.

REVIEWERS' COMMENTS:

Reviewer #1 (Remarks to the Author):

The authors have nicely responded to my comments.

My main remaining concern is related to the fact that they are simply reporting correlations and there is no proof that the detected methylation changes are in fact caused by the inversions. They state this in several places in the manuscript such as in line 282: It should be noted that there are four genes (KANSL1, MAT, LOC100128977, and WNT3) in this region with significant associations that were also differentially methylated by the inversion.

Reviewer #2 (Remarks to the Author):

The authors have addressed the concerns raised in the review with the analyses and explanations they have added, so I recommend the article to be accepted for publication.

Reviewer #3 (Remarks to the Author):

The authors have addressed the concerns raised in the review with the analyses and explanations they have added, so I recommend the article to be accepted for publication.

Reviewer #1 (Remarks to the Author):

The authors have nicely responded to my comments. My main remaining concern is related to the fact that they are simply reporting correlations and there is no proof that the detected methylation changes are in fact caused by the inversions. They state this in several places in the manuscript such as in line 282: It should be noted that there are four genes (KANSL1, MAT, LOC100128977, and WNT3) in this region with significant associations that were also differentially methylated by the inversion.

Response: We agree that it is more precise to tone down the sentences that suggest a causal relationship of the inversion. To address this issue, we have replaced the text “differentially methylated by the inversion” by “differentially methylated according to the inversion haplotype” or “differentially methylated depending on the inversion haplotype”.

These changes can be found at lines 157, 172, 238, 240, 248, 255, 273, 406, and in Figure 1 caption.

Here we show an example of the change in the text highlighted by the reviewer (line 253):

“It should be noted that there are four genes (KANSL1, MAT, LOC100128977, and WNT3) in this region with significant associations that were also differentially methylated depending on the inversion haplotype.”

Reviewers #2-3 (Remarks to the Author):

The authors have addressed the concerns raised in the review with the analyses and explanations they have added, so I recommend the article to be accepted for publication.

Response: We thank the reviewers for their comments and their recommendation to accept our paper for publication.